# Changes in loss to follow-up among elderly patients with schizophrenia during the COVID-19 pandemic: A study from a specialized center in Peru

Paulo Ruiz-Grosso[1,2*], Luis Macedo-Orrego[2,3], Sonia Zevallos-Bustamante[2]

**1** Facultad de Medicina Alberto Hurtado, Universidad Peruana Cayetano Heredia, Lima, Perú, **2** Dirección Ejecutiva Investigación, Docencia y Atención Especializada de Adultos y Adultos Mayores, Instituto Nacional de Salud Mental "Honorio Delgado - Hideyo Noguchi", Lima, Perú, **3** Facultad de Medicina, Universidad Nacional Mayor de San Marcos, Lima, Perú

* paulo.ruiz@upch.pe

## Abstract

To estimate the survival function and identify risk factors for loss to follow-up (LTFU) among elderly patients with schizophrenia treated at the National Institute of Mental Health (INSM) in Peru. A retrospective cohort study was conducted using clinical records from 138 randomly selected patients diagnosed with schizophrenia. Eligible patients had attended at least one appointment between April 15, 2018, and April 15, 2022. Clinical and attendance data were recorded for each psychiatric appointment until the end of the follow-up period or LTFU. The survival function for LTFU was estimated, and associations with type of appointment (telemedicine vs. face-to-face) and period of first appointment (pre- vs. post-telemedicine implementation) were analyzed using Cox regression. The overall survival function at the end of the follow-up period was 47%. Patients whose treatment began at INSM after the implementation of telemedicine (TM) had a survival function of 49%, compared with 75% for those who initiated their treatment before TM. LTFU was significantly associated with initiating treatment in the Emergency Department (ED) rather than through outpatient care and with starting treatment post-COVID-19. Our findings indicate a higher rate of LTFU among patients who began treatment at INSM during the post COVID-19 period or through the Emergency Department. These results highlight the need for targeted interventions to address these risk factors and improve continuity of care for this vulnerable population.

## Introduction

The management of the COVID-19 pandemic, including the continuity of care for chronic conditions such as schizophrenia and other mental disorders, represented a

**Data availability statement:** The minimal de-identified dataset and analysis code supporting this study are openly available in Zenodo at https://doi.org/10.5281/zenodo.17536051.

**Funding:** The authors received no specific funding for this work.

**Competing interests:** The authors have declared that no competing interests exist.

significant global challenge" [1–3]. In 2022, the World Health Organization reported early evidence of the pandemic's impact on mental health, noting increases in depression and anxiety and a heightened risk of severe illness or death among individuals with pre-existing mental disorders. However, the evidence did not support a corresponding increase in suicide rates [4].

During the COVID-19 pandemic, older adults were at higher risk of developing more severe forms of disease and experiencing higher mortality, as well as increased vulnerability to depressive and anxiety symptoms. Female sex, younger age groups within the elderly population, lower income and educational attainment, and the presence of pre-existing psychiatric disorders were identified as factors associated with higher levels of anxiety and depression during this period [5]. Evidence regarding the impact of COVID-19 on chronic psychotic disorders such as schizophrenia in older adults is limited. However, available data suggest that social isolation, the loss of close relatives, and economic challenges—documented among younger populations—may also have contributed to the perceived increase in psychotic symptoms among older adults. This association may have been confounded by the mismanagement of other chronic conditions, the use of corticosteroids in severe COVID-19 cases or even reports of a slight increase in psychotic symptoms following COVID-19 vaccination, particularly among individuals hesitant to be vaccinated [6–9].

Peru was one of the countries most severely affected by the COVID-19 pandemic worldwide. According to the Johns Hopkins Coronavirus Resource Center, Peru reported a mortality rate of 665 deaths per 100,000 population—the highest globally—and ranked fifth worldwide for case fatality rate, with 4.9% of confirmed cases resulting in death [10]. In addition, local studies documented an increase in symptoms of depression and anxiety, along with a reduction in mental health consultations at primary care centers during the pandemic [11,12].

During the COVID-19 pandemic, telemedicine rapidly emerged as a vital tool within Peru's healthcare system. Although a regulatory framework for telehealth had been established in 2005, the pandemic catalyzed a major expansion of these services to meet urgent healthcare demands [13]. Telemedicine played a key role in maintaining access to care, particularly for mental health outpatient services at major national referral centers, despite significant infrastructural limitations such as uneven access to high-speed internet and the lack of interoperable health information systems [14].

Loss to follow-up (LTFU) in patients with schizophrenia is a critical concern, as it has been reported in 10–62% of patients [15–17]. The discontinuation of treatment has been associated with an increased risk of psychotic relapses, hospitalization, increased use of emergency services, and, as a consequence of relapse, may also face increased risk of unemployment and housing instability [15,16,18–20]. Although no studies have specifically examined the impact of LTFU on treatment outcomes in older adults, older age has been associated with an increased likelihood of LTFU [16].

A preliminary study conducted at the National Institute of Mental Health (INSM), Peru, using early longitudinal data, reported that patients receiving treatment in the

psychosis and elderly programs (aged 60 years and older) had a lower risk of loss to follow-up (LTFU) compared with those treated for mood, anxiety, or personality disorders. The data also indicated that patients whose first appointment occurred during the COVID-19 pandemic had a higher risk of LTFU than those whose initial visit took place before the pandemic.

Despite the profound impact of COVID-19 on mental health services, there is limited evidence on how these disruptions affected continuity of care, particularly among older adults with chronic psychotic disorders such as schizophrenia. Furthermore, the extent to which alternative care delivery strategies adopted during the pandemic—such as telemedicine—mitigated or exacerbated LTFU in this population remains unclear. This study seeks to address this gap by examining patterns and predictors of follow-up discontinuation among older adults with schizophrenia treated at a national referral center in Peru.

## Methods

### Study design

We conducted a retrospective cohort study of patients aged 60 years and older who were diagnosed with schizophrenia (ICD-10 codes F20.0–F20.9) and who had at least one consultation in the Emergency Department or an ambulatory service at the Instituto Nacional de Salud Mental Honorio Delgado–Hideyo Noguchi (INSM) between April 15, 2018, and April 15, 2022. The follow-up period allowed monitoring of individuals for at least two years and included both those who began care at INSM in the two years preceding the introduction of telemedicine—implemented in response to the COVID-19 pandemic on April 15, 2020—and those whose first consultation occurred thereafter. The primary outcome was loss to follow-up (LTFU), defined as discontinuation of scheduled consultations in the absence of documented discharge, transfer to another healthcare provider, or additional information confirming subsequent follow-up.

### Sample and sampling

The target population consisted of patients aged 60 years and older with a diagnosis of schizophrenia who received care during the two years surrounding the onset of the COVID-19 pandemic. The sampling frame included all patients aged 60 years and older who attended at least one Emergency Department or ambulatory consultation at the INSM between April 15, 2018, and April 15, 2022.

The INSM is a national referral center for mental health care that supports first-, second-, and third-level health services in northern Lima. Its scope includes outpatient care for individuals referred after the establishment of Community Mental Health Centers (CMHCs), as well as patients who began treatment before this referral system and who remain under INSM care. The institute also provides hospitalization, rehabilitation, and emergency services. From the sampling frame, 150 participants were randomly selected for this preliminary study.

At the onset of the COVID-19 pandemic, first-level healthcare services, including CMHCs, were disrupted, as most resources were redirected to COVID-19 care. This resulted in a temporary decline in mental health consultations, which returned to pre-pandemic levels after approximately one year, as reported by Diez-Canseco et al. This disruption may limit the generalizability of our findings to populations initiating care after the pandemic and during the implementation of telemedicine.

### Study procedures

Data were collected using the Google Forms application, which included sociodemographic information (e.g., date of birth, sex) as well as details about each consultation at the Emergency Department and with the treating psychiatrist. This data included the date of the appointment, type of consultation (telemedicine vs. face-to-face), specific diagnosis, and other relevant data. Access to the dataset was restricted to the study authors.

Data were obtained from both physical and electronic medical records. This dual approach was necessary because information from the pre-pandemic period, as well as Emergency Department consultations during both pre- and post-pandemic periods, was available only in physical records.

### Study variables

The primary outcome variable was loss to follow-up (LTFU), defined as the discontinuation of scheduled appointments after at least one consultation (in the Emergency Department, via telemedicine, or through face-to-face ambulatory care), without documentation in the medical chart of referral to another institution or death. For example, if a patient attended an Emergency Department consultation and the clinical chart indicated referral to a Community Mental Health Center (CMHC), this case was not classified as LTFU.

Covariates included sociodemographic variables (age, sex, education, and marital status) recorded in the clinical chart. Clinical variables comprised the duration of care at INSM at the start of the follow-up period, schizophrenia subtype diagnosis, other psychiatric and non-psychiatric comorbidities, prescribed medications, and the date of the first consultation at INSM. Special attention was given to the type of consultation (telemedicine vs. in-person), categorized as occurring before or after April 15, 2020—the date telemedicine was introduced in response to the COVID-19 pandemic in Peru.

### Data processing and statistical analysis

Data analysis followed a structured process, beginning with the exploration and description of individual variables, followed by estimation of the survival function, bivariate analyses, and multivariable analyses with diagnostic procedures to assess model assumptions. Descriptive analyses were performed using measures of central tendency and dispersion.

The survival function for the primary outcome variable, loss to follow-up (LTFU), was estimated using time-to-event methods, including Kaplan–Meier survival tables and graphical representations. Bivariate and multivariable associations were examined using Cox regression models. All statistical analyses were conducted using Stata software, version 18.

### Sample size and statistical power

Sample size for the full study was estimated assuming a Type I error of 5% and a Type II error of 20%, with a minimum detectable difference of 15% in the incidence of events considered significant. Based on these parameters, a total of 375 individuals was determined to be necessary.

### Ethics statement

This study received approval from the Institutional Review Board of Universidad Peruana Cayetano Heredia on January 10, 2023 (Approval Number: 018-02-23; Code: 209992). The requirement for informed consent was waived as part of the approved protocol. Data were accessed for research purposes between August 12 and November 30, 2023. Personally identifiable information was stored in a separate database, which was linked to the clinical and sociodemographic data through anonymized alphanumeric codes. Only the principal investigator had access to the identity database.

## Results

### Data description

Data from 1,378 consultations, corresponding to 138 individuals, were collected. The mean age of participants was 69.6 (SD = 8.0), and 53.9% of participants were female. The most common education level attained was complete elementary school (38.8%), followed by complete secondary school (30.2%), superior education (16.6%) and less than elementary school (14.4%). The most common marital status was single (56.1%), followed by married or cohabitant (36%), and separated, divorced, or widowed (7.9%).

Regarding clinical variables, 59 patients (55.7%) had their first consultation at INSM through in-person ambulatory appointments, 45 (42.4%) through Emergency Department visits, and 2 (1.9%) through ambulatory telemedicine appointments. During the follow-up period, 519 consultations (38%) were in-person ambulatory, 598 (44%) were telemedicine ambulatory, and 249 (18%) were Emergency Department visits.

Concerning the onset of telemedicine use at INSM and the COVID-19 pandemic, 99 patients (76.1%) had their first appointment before the pandemic and 31 (23.9%) afterwards. Overall, 610 consultations (44.3%) occurred before the introduction of telemedicine, and 768 (55.7%) after its implementation. The mean number of visits during the follow-up period was 17.3 (SD = 9.5); 27 individuals (19.6%) attended only one appointment, and 47 (34.1%) attended three or fewer. The most common diagnoses were schizophrenia (56.8%), other psychoses (18.0%), and affective disorders (6.3%). Loss to follow-up (LTFU) was observed in 61 individuals (44.9%). The mean psychiatric drug use score for the overall sample was 102.9 (SD = 64.3) units. Further details of the sample are presented in Table 1. For the estimation of the survival function, data from 111 participants were analyzed. The survival function for the full sample was 47% (95% CI: 31–61%). When stratified by the period of first attention at INSM, we found that the survivor function for those that started their attention at the INSM before the implementation of telemedicine was of 74% (95% CI: 63–82%), while for those that started afterwards was of 49% (95% CI: 24–71%), for a comparable 2-year period of follow up. The survivor function for those whose first attention was in an ambulatory appointment was 64% (95% CI: 40–80%), while for those initiated care in the emergency department it was 22% (95% CI: 7–44%). Further details are presented in Figs 1 and 2 and in Table 2.

## Bivariate analysis

Cox regression analysis showed that having the first consultation at INSM in the Emergency Department (HR = 4.2; 95% CI: 2.0–8.7), receiving care after the implementation of telemedicine (HR = 2.1; 95% CI: 1.0–4.3), Emergency Department visits during follow-up (HR = 3.2; 95% CI: 1.7–7.2), and initiating treatment at INSM after the implementation of telemedicine (HR = 3.9; 95% CI: 1.8–8.7) were all associated with a higher hazard of loss to follow-up (LTFU). In contrast, a greater number of follow-up visits was associated with a lower risk of LTFU (HR = 0.8; 95% CI: 0.7–0.8). Further details are presented in Table 3. The variables *type of first consultation at INSM* (ambulatory vs. Emergency Department), *number of consultations*, and *age* did not satisfy the proportional hazards assumption ($p < 0.05$ in the proportional hazards test using the post-estimation phtest command in Stata). Therefore, we fitted Cox regression models allowing for time-varying effects to account for changes in the hazard over follow-up. These results showed that the average hazard during follow-up remained significant for type of first consultation at INSM (HR = 4.57; 95% CI: 2.20–9.50), while no evidence of statistical significance was found for its interaction with time (Coef = 1.39; 95% CI: 0.84–2.28). Age was not associated with LTFU on average (HR = 0.97; 95% CI: 0.92–1.02); however, its interaction with time was significant, indicating an increasing effect across follow-up (Coef = 0.98; 95% CI: 0.96–0.99). These results are summarized in Table 3, and robust variance estimates were applied to these models.

As a sensitivity analysis, we fitted bivariate Weibull accelerated failure time models. Admission through the Emergency Department was associated with shorter time to LTFU (Coef = –1.56; 95% CI: –2.47 to –0.65), with no evidence of variation in the hazard across follow-up (shape parameter = 0.84; 95% CI: 0.61–1.15). Age showed no effect (Coef = 0.002; 95% CI: –0.08 to 0.08), but there was evidence of a decreasing hazard over time (shape parameter = 0.60; 95% CI: 0.45–0.79).

## Multivariate and exploratory analyses

**Multivariate models.** Based on the results of the bivariate analyses, we developed two separate multivariable models. In the first model, the primary predictor of LTFU was the period of the first consultation at INSM (before vs. after the implementation of telemedicine). In the second model, the primary predictor was the type of first consultation at INSM (ambulatory vs. Emergency Department). The decision to present two separate models was based on the

**Table 1. Sample characteristics.**

| Variable | n | % |
|---|---|---|
| *Sex* | | |
| Female | 75 | 53.9 |
| Male | 64 | 46.1 |
| *Age* (mean, sd) | 69.5 | 0.68 |
| *Marriage status* | | |
| Single | 78 | 56.1 |
| Married or cohabitant | 50 | 36 |
| Divorced, separated or widow | 11 | 7.9 |
| *Education* | | |
| Less than elementary | 20 | 14.4 |
| Elementary School | 54 | 38.8 |
| Secondary School | 42 | 30.2 |
| Superior education | 23 | 16.6 |
| *Type of first consultation at INSM* | | |
| Ambulatory, face to face | 61 | 57.5 |
| Emergency department | 45 | 47.5 |
| *Tipe of consultation* | | |
| Ambulatory, face to face | 518 | 37.9 |
| Ambulatory, telemedicine | 598 | 43.8 |
| Emergency department | 249 | 18.3 |
| *Period of first attention at INSM* | | |
| Pre-COVID | 99 | 76.1 |
| Post-COVID | 31 | 23.9 |
| *Period of visit* | | |
| Pre-COVID | 610 | 44.3 |
| Post-COVID | 768 | 55.7 |
| Number of visits during follow up (mean, SD) | 17.3 | 9.5 |
| *Diagnosis* | | |
| *Schizophrenias* | 63 | 56.76 |
| Other psychoses | 20 | 18.02 |
| Affective disorders* | 7 | 6.3 |
| Other diagnoses** | 21 | 18.9 |
| Drug score (mean, sd) | 102.8 | 73 |
| *Loss to follow up* | | |
| Yes | 75 | 55.1 |
| No | 61 | 44.9 |

\* Depressive, recurrent depressive and bipolar disorders.

\*\* Anxiety, dementia, addiction and other disorders.

fact that, due to procedural changes at INSM during the COVID-19 pandemic, individuals could not have their first consultation as ambulatory patients in the post-telemedicine implementation period. Including both variables in the same model could therefore have created difficulties in interpretation. To avoid overadjustment, we used Directed Acyclic Graphs (DAGs) to determine which covariates should be included in each model. These DAGs are provided in Figs 3 and 4.

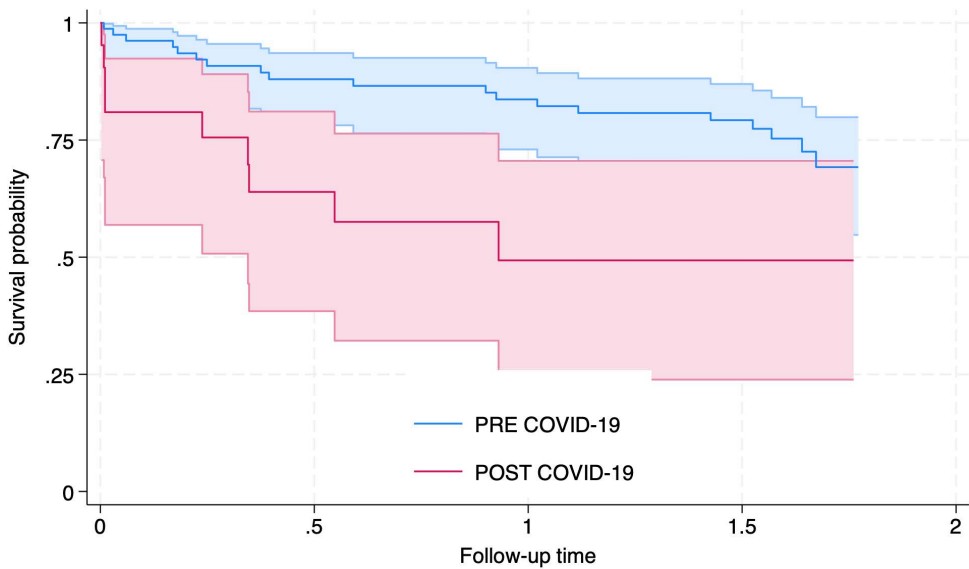

**Fig 1. Kaplan–Meier estimates of time to loss to follow-up (LTFU) according to Period of First Attention (PRE/POST COVID-19).** Shaded areas represent 95% confidence intervals.

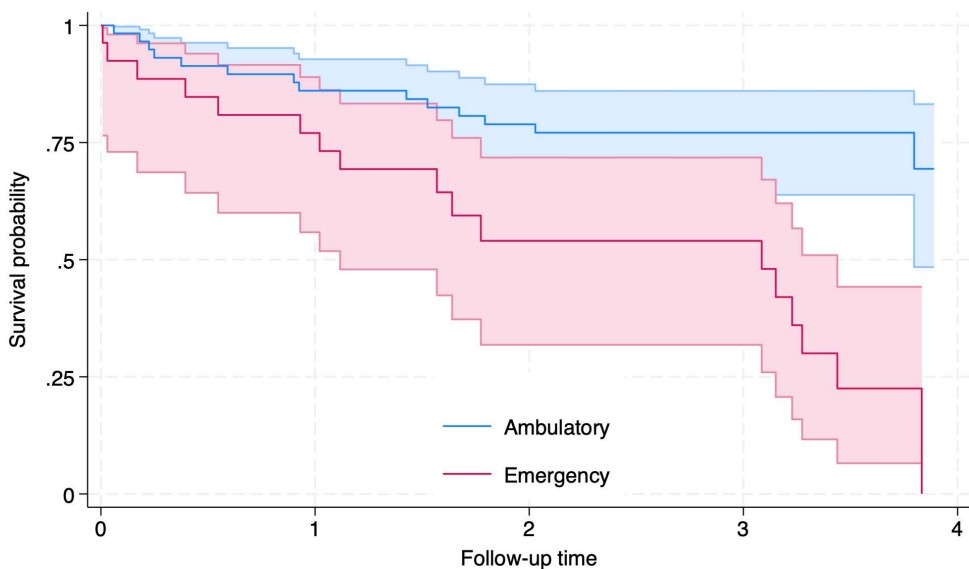

**Fig 2. Kaplan–Meier estimates of time to loss to follow-up (LTFU) according to Type of First Attention (Ambulatory/Emergency Department).** Shaded areas represent 95% confidence intervals.

In the first model, the effect size of period of first consultation at INSM did not remain statistically significative, even if the strength of association was nearly unchanged (HR = 2.96, 95% CI: 0.96–9.11, p = 0.058). No covariate was found significatively associated to LFTU in this model.

The second model showed that patients that had their first attention at the emergency department had nearly three-fold higher IR of treatment dropout compared with the reference group (HR = 2.99, 95% CI: 1.26–7.09, p = 0.013). The

**Table 2. Survivor function for LTFU by type of admission, and period of initiation of care at INSM.**

| | Time | At Risk | Fail | Net lost | Survivor function | 95%CI | At Risk | Fail | Net lost | Survivor function | 95%CI |
|---|---|---|---|---|---|---|---|---|---|---|---|
| | | *Ambulatory patients* | | | | | *Emergency departmen* | | | | |
| Type of Admission | 0 | 0 | 5 | -57 | 1.00 | | 0 | 4 | -26 | 1.00 | |
| | 0.5 | 52 | 3 | 0 | 0.91 | 0.80–0.96 | 22 | 2 | 0 | 0.85 | 0.64–0.94 |
| | 1 | 49 | 1 | 1 | 0.86 | 0.74–0.93 | 20 | 2 | 4 | 0.77 | 0.56–0.89 |
| | 1.5 | 47 | 3 | 0 | 0.84 | 0.72–0.92 | 14 | 3 | 1 | 0.69 | 0.48–0.83 |
| | *2* | *44* | *1* | *1* | *0.79* | *0.66–0.87* | *10* | *0* | *1* | *0.54* | *0.32–0.72* |
| | 2.5 | 42 | 0 | 3 | 0.77 | 0.64–0.86 | 9 | 0 | 0 | 0.54 | 0.32–0.72 |
| | 3 | 39 | 0 | 6 | 0.77 | 0.64–0.86 | 9 | 5 | 2 | 0.54 | 0.32–0.72 |
| | 3.5 | 33 | 2 | 26 | 0.77 | 0.64–0.86 | 2 | 1 | 1 | 0.23 | 0.07–0.44 |
| | 3.9 | 5 | 0 | 5 | 0.64 | 0.40–0.80 | 0 | 0 | 0 | . | |
| | | *PRE-Telemedicine* | | | | | *Post-Telemedicine* | | | | |
| Period of first admission | 0 | 0 | 9 | -77 | 1.00 | | 0 | 7 | -18 | 1.00 | |
| | 0.5 | 68 | 3 | 0 | 0.89 | 0.79–0.94 | 11 | 2 | 3 | 0.64 | 0.38–0.81 |
| | 1 | 65 | 3 | 1 | 0.85 | 0.74–0.91 | 6 | 0 | 4 | 0.49 | 0.24–0.71 |
| | 1.5 | 61 | 5 | 1 | 0.81 | 0.70–0.88 | 2 | 0 | 1 | 0.49 | 0.24–0.71 |
| | 2 | *55* | *1* | *2* | *0.74* | *0.63–0.82* | *1* | *1* | *0* | *0.49* | *0.24–0.71* |
| | 2.5 | 52 | 0 | 3 | 0.73 | 0.61–0.81 | 0 | 0 | 0 | . | |
| | 3 | 49 | 5 | 8 | 0.73 | 0.61–0.81 | 0 | 0 | 0 | . | |
| | 3.5 | 36 | 3 | 27 | 0.65 | 0.53–0.75 | 0 | 0 | 0 | . | |
| | 3.9 | 6 | 0 | 6 | 0.51 | 0.33–0.66 | 0 | 0 | 0 | . | |

interaction of this variable with time was not statistically significant (HR = 1.38, 95% CI: 0.82–2.34, p = 0.226), suggesting that the effect of emergency admission remained stable across the follow-up period. In contrast, age showed evidence of a time-varying effect: although the average effect was not significant (HR = 0.95, 95% CI: 0.89–1.02, p = 0.151), the interaction with log(time) indicated that the protective association of older age increased over time (HR for interaction = 0.96, 95% CI: 0.93–1.00, p = 0.046). Other covariates, including education level and psychiatric diagnosis, were not significantly associated with dropout IR.

As a sensitivity analysis, we fitted an AFT model with a Weibull distribution. Results were consistent with the multivariate models of the Cox regression. First attention at the emergency department was associated with a significantly shorter time to LTFU (coef = −1.08, p = 0.032), indicating accelerated discontinuation. Age showed a non-significant tendency toward longer retention, and no significant associations were observed for education or psychiatric diagnosis. The shape parameter (p = 0.86, 95%CI:0.62-1.20) suggested a decreasing hazard of dropout over time.

**Exploratory analyses.** In the first exploratory analysis, the association between the type of first consultation at INSM (ambulatory vs. Emergency Department) and LTFU was estimated only among patients who initiated care before the implementation of telemedicine. In this subgroup, initiating care through the Emergency Department was significantly associated with LTFU in the bivariate analysis (HR = 4.42; 95% CI: 2.29–8.89), and remained strong, although not statistically significant, in the adjusted model (HR = 2.87; 95% CI: 0.90–9.14).The second exploratory analysis examined the association between the average drug score during follow-up and LTFU. Compared with patients in the lowest quartile of drug score, those in the second quartile had a significantly lower hazard of LTFU (HR = 0.39; 95% CI: 0.15–0.99). A similar magnitude of association was observed in the highest quartile, although it did not reach statistical significance (HR = 0.33; 95% CI: 0.11–1.04).

**Table 3. Bivariate and multivariate analysis.**

| Variable | Bivariate | | | Multivariate*** | | | Multivariate**** | | |
|---|---|---|---|---|---|---|---|---|---|
| | OR | p value | 95%CI | OR | p value | 95%CI | OR | p value | 95%CI |
| *Sex* | | | | | | | | | |
| Female | 1 | – | | 1 | – | | | | |
| Male | 1.2 | 0.556 | 0.65–2.22 | 1.2 | 0.645 | 0.58–2.38 | | | |
| *Age* (mean, sd) | 1 | 0.97 | 0.92–1.02 | | | | 0.95 | 0.241 | 0.89–1.02 |
| *Marriage status* | | | | | | | | | |
| Single | 1 | – | | 1 | – | | | | |
| Married or cohabitant | 1.4 | 0.291 | 0.74–2.69 | 1.4 | 0.377 | 0.67–2.90 | | | |
| Divorced, separated or widow | 1.3 | 0.685 | 0.38–4.31 | 1.6 | 0.488 | 0.43–5.97 | | | |
| *Education* | | | | | | | | | |
| Less than elementary | 1 | – | | | | | | | |
| Elementary School | 1.1 | 0.91 | 0.35–3.22 | 1.4 | 0.558 | 0.44–4.58 | 0.76 | 0.687 | 0.19–2.95 |
| Secondary School | 2.3 | 0.143 | 0.76–6.88 | 2.4 | 0.143 | 0.75–7.47 | 1.5 | 0.533 | 0.42–5.33 |
| Superior education | 1.1 | 0.843 | 0.32–4.04 | 1.4 | 0.652 | 0.35–5.43 | 1.16 | 0.848 | 0.26–5.25 |
| *Type of first consultation at INSM* | | | | | | | | | |
| Ambulatory | 1 | – | | | | | 1 | – | |
| Emergency department | **4.2** | **<0.001** | **2.06–8.45** | | | | **2.99** | **0.013** | **1.26–7.08** |
| *Tipe of consultation* | | | | | | | | | |
| Ambulatory, face to face | 1 | – | | | | | | | |
| Ambulatory, telemedicine | 1.4 | 0.415 | 0.60–3.49 | | | | | | |
| Emergency department | **3.2** | **0.004** | **1.47–7.14** | | | | | | |
| *Period of first attention at INSM* | | | | | | | | | |
| Pre-COVID | 1 | – | | 1 | – | | | | |
| Post-COVID | **3.9** | **0.001** | **1.76–8.66** | 3 | 0.058 | 0.96–9.12 | | | |
| *Period of visit* | | | | | | | | | |
| Pre-COVID | 1 | – | | 1 | – | | | | |
| Post-COVID | **2.1** | **0.041** | **1.03–4.30** | 1.13 | 0.810 | 0.41–3.10 | | | |
| Number of visits during follow up (mean, SD) | **0.8** | **<0.001** | **0.72–0.82** | | | | | | |
| *Diagnosis* | | | | | | | | | |
| Schizophrenias | 1 | – | | 1 | – | | 1 | – | |
| Other psychoses | 1.5 | 0.536 | 0.44–4.94 | 1.5 | 0.528 | 0.41–5.64 | 1.86 | 0.472 | 0.34–10.01 |
| Affective disorders* | 2.3 | 0.073 | 0.92–5.72 | 1.9 | 0.215 | 0.69–5.21 | 1.73 | 0.392 | 0.49–6.04 |
| Other diagnoses** | 1.7 | 0.209 | 0.75–3.80 | 1.6 | 0.290 | 0.67–3.85 | 1.93 | 0.189 | 0.72–5.14 |
| Drug score (mean, sd) | 1 | 0.073 | 0.99–1.00 | | | | | | |

\* Depressive, recurrent depressive and bipolar disorders.

\*\* Anxiety, dementia, addiction and other disorders.

\*\*\* Multivariate model for period of first attention at INSM as main independent variable.

\*\*\*\* Multivariate model for type of consultation at first attention at INSM as main independent variable.

## Discussion

The overall survival function for LTFU in the entire sample was 47%. When examining survival estimates after two years of follow-up, individuals who initiated care at INSM following the implementation of telemedicine had a survival rate of 49%, whereas those who began care before this period showed a significantly higher estimate of 82% (see Table 2). Exploratory bivariate analyses indicated that both initial and follow-up consultations at the Emergency Department—compared

PLOS Mental Health

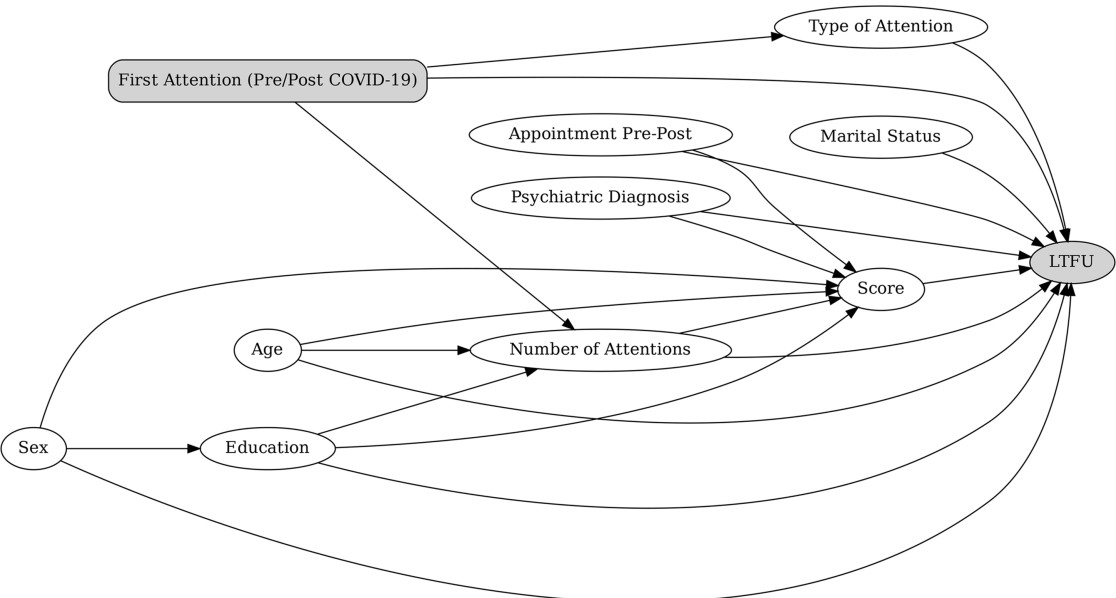

**Fig 3. Directed acyclic graph (DAG) of hypothesized relationships between patient characteristics and loss to follow-up (LTFU), with Period of First Attention (Pre/Post COVID-19) as the exposure.**

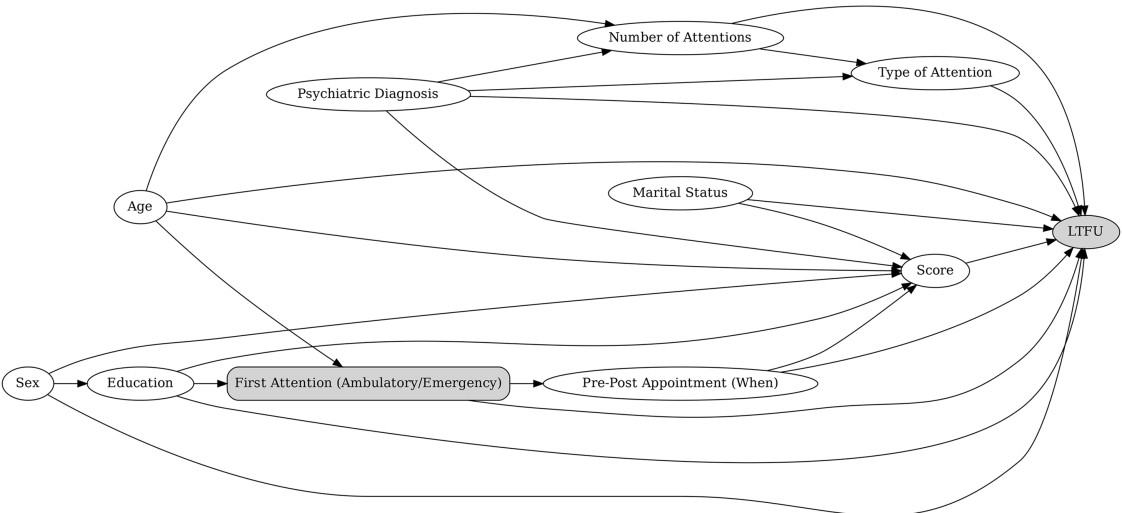

**Fig 4. Directed acyclic graph (DAG) of hypothesized relationships between patient characteristics and loss to follow-up (LTFU), with Type of First Attention (Ambulatory/Emergency) as the exposure.**

to in-person ambulatory appointments—were associated with a higher IR of LTFU. In multivariate models, only initial consultations at the Emergency Department remained statistically significant.

Before interpreting these results, some limitations must be acknowledged. Of particular importance is the definition of the primary outcome, LTFU. In this study, LTFU was operationalized as the absence of follow-up appointments after at least one consultation with a psychiatrist. Manual reviews of clinical records were conducted to identify and exclude cases

with documented referral to another health facility or death; these cases were not considered LTFU. However, it was not possible to determine the exact reasons for LTFU for all individuals in the sample, as health facilities in Peru do not share a unified information system. Some individuals classified as LTFU may have been referred to primary care settings without documentation in the clinical chart, may have died during follow-up without notification to INSM, or may have experienced transient symptoms—particularly those with only one or two emergency visits—and therefore did not continue treatment. A final assessment of the clinical or vital status of individuals lost to follow-up was beyond the scope of this study, as the necessary data sources were not available.

The impact of this potential misclassification bias would depend on the distribution of the underlying causes of LTFU. For example, if a large proportion of individuals had only a few emergency visits and subsequently improved, or if some attended the emergency department to continue pharmacological treatment because primary care CMHCs in their communities were closed during the first year of the pandemic, these cases may have been misclassified as LTFU. Such misclassification could disproportionately increase the number of individuals categorized as LTFU in the year following the onset of telemedicine and, in turn, may have biased the estimates toward rejecting the null hypotheses on both bivariate and multivariate models.

Regarding external validity, the findings are most applicable to individuals aged 60 years or older receiving care at a specialized fourth-level referral center serving the northern region of Lima. Given the chronic nature of schizophrenia and the participants' age, it is likely that many initiated treatments prior to the implementation of the decentralized mental health care model introduced over the past decade. This could explain why they remained under care at the referral center instead of being transitioned to first-level institutions, as intended by policy. Furthermore, during the first year of the COVID-19 pandemic, disruptions at the first level of mental healthcare significantly affected Community Mental Health Centers (CMHCs), ranging from complete closures to partial service interruptions [12]. These circumstances may have influenced the profile of patients seeking care at INSM, which maintained continuous Emergency Department operations and experienced minimal service disruption during the early phase of the pandemic. Finally, although participants were randomly selected from a list of eligible individuals, the relatively small sample size limited the statistical power to detect anything but large differences.

Finally, another important limitation is that the categories defining the period of first contact at INSM (pre- and post-telemedicine implementation) closely overlapped with the onset of the COVID-19 pandemic in Peru and the systemic changes it produced. These included the partial or total closure of CMHCs, increased demand for mental health services, and reduced availability of care in non-specialized settings. The pandemic was officially declared in Peru on March 6, 2020, and telemedicine was implemented at INSM on April 15, 2020. From that point, telemedicine became the only available modality for ambulatory care, precluding face-to-face consultations. Therefore, the variable "period of first contact" should be interpreted as a partial proxy for broader systemic changes that are difficult to discern with the available data. Moreover, this variable does not perfectly capture the type of care received, as face-to-face visits were gradually reintroduced about one year later. Thus, patients initiating care in the "post-telemedicine" period may have received both telemedicine and face-to-face consultations during follow-up, unless they were LTFU earlier.

Despite these limitations, several key findings of the study merit attention. First, the survival function was notably higher at both one year and approximately two years of follow-up among individuals who initiated care at INSM prior to the implementation of telemedicine during the onset of the COVID-19 pandemic in Peru. This group also demonstrated a lower IR of LTFU in bivariate analyses. Several factors may account for these observations. As previously discussed, the onset of the pandemic led to a significant disruption in first-level mental health services in Peru, which only began to recover after approximately one year [12]. As a result, some patients may have sought care at the INSM Emergency Department due to acute symptom exacerbations, potentially related to treatment discontinuation or medication shortages, or simply as a means to access necessary psychiatric care.

As first-level mental health care services resumed, it is plausible that some patients re-engaged with their local Community Mental Health Centers (CMHCs) or general hospitals, where care could continue without requiring a formal referral from INSM. However, such transitions are frequently undocumented in clinical records, limiting the ability to capture evidence of this normalization of care. This limitation may also account for the more pronounced decline in the survival function observed after one year of follow-up among individuals who initiated treatment at INSM following the implementation of telemedicine during the pandemic.

Another possible explanation is that some individuals may have experienced de novo psychotic symptoms triggered by factors such as the use of certain medications—particularly corticosteroids—or as a direct consequence of COVID-19 infection. Previous studies have reported an increase in psychotic symptoms, behavioral disturbances, and aggressive behavior among the elderly during the COVID-19 pandemic, as well as an increase in the average number of psychiatric diagnoses in specialized care centers [6,21,22]. However, these presentations may not have been accurately classified at the time, and broader or non-specific diagnostic labels could have been applied. In such cases, it is plausible that individuals experienced a rapid resolution of symptoms, leading patients or their families to perceive continued care as unnecessary. Alternatively, some may have died due to COVID-19 or other causes and consequently did not return for follow-up.

Second, in the exploratory multivariate analyses, after controlling for potential confounders, only the type of initial contact at INSM remained significantly associated with LTFU, specifically among individuals whose first consultation occurred in the Emergency Department. This finding may reflect the differing circumstances under which patients accessed emergency versus ambulatory care, both before and after the implementation of telemedicine. Prior to the pandemic, ambulatory care at INSM was generally initiated through referrals from lower-level mental health services. However, during the first year of the pandemic, with reduced availability of services at CMHCs, some patients accessed care directly through the Emergency Department. As previously discussed, the disruption of first-level mental health services likely contributed to increased Emergency Department visits due to medication shortages or acute decompensation among previously stable patients.

Moreover, exploratory analyses indicated that, even among individuals who initiated care at INSM before the implementation of telemedicine, beginning treatment through the Emergency Department was associated with a higher IR of LTFU. Although this association lost statistical significance after adjusting for confounders, it suggests that patients initiating care via emergency services may differ in clinical or sociodemographic profile from those beginning care through ambulatory services in non-pandemic contexts. For instance, a study by Padilha et al. on psychiatric emergencies in a Brazilian general hospital found that the most common reasons for consultation were depressive symptoms, agitation, and psychoactive substance-related issues [23]. Similarly, in a study of elderly patients in southern India, Mukku et al. reported a predominance of male patients, with agitation, confusion, and risk of self-harm being the main reasons for emergency visits; nearly one-third of these patients were diagnosed with neuropsychiatric conditions [24].

Thus, it can be speculated that if approximately one-third of older adults receiving care in the Emergency Department present with neuropsychiatric conditions, it is plausible that, once symptoms such as agitation or confusion are stabilized, they may continue treatment in non-psychiatric healthcare settings. In contrast, older adults whose first consultation occurs in the ambulatory services at INSM are more likely to include individuals with chronic psychotic disorders who have been receiving treatment for many years—sometimes for over three decades—or new patients formally referred by psychiatrists from lower-level facilities, such as CMHCs or general hospitals. This interpretation, however, would benefit from further research to more precisely describe and compare the clinical and sociodemographic profiles of older adults initiating care in the Emergency Department, both during and outside of pandemic periods.

Our findings underscore the importance of addressing LTFU among older adults with schizophrenia, particularly those who initiate care in emergency settings. Early alert systems to detect individuals at risk of LTFU have shown promise and may be relatively easy to implement in both specialized and primary care settings. Such systems can include preventive actions such as text-based reminders, open communication channels that facilitate easy contact with health providers,

active outreach to patients who miss appointments, and automated programs that track key variables through information recorded by mobile devices, which are increasingly accessible even in low-income settings. [25–27]. The implementation of structured follow-up protocols that utilize hybrid care models—integrating telemedicine with face-to-face appointments—may improve continuity and sustainability of care, especially in contexts with limited human resources [28,29]. Finally, healthcare system leadership should prioritize the development of interoperable digital health infrastructures and governance frameworks that ensure continuity-of-care information is shared seamlessly across primary and specialized services, thereby strengthening coordination and reducing LTFU.

From a theoretical standpoint, both the results and limitations of our study reinforce the notion that LTFU in psychiatric care should be conceptualized as a multidimensional construct rather than a simple indicator of patient disengagement. In our sample, this construct may encompass cohort-specific factors, such as systemic disruptions during the COVID-19 pandemic, as well as more common events, including undocumented transitions to primary or private care, deaths occurring outside the healthcare system, or the natural resolution of acute presentations—particularly among patients initiating care through emergency services. These findings also underscore the role of emergency services as entry points within continuity-of-care frameworks. More broadly, this research highlights the need for theoretical models of adherence and continuity in mental health to integrate both individual-level determinants and system-level dynamics, especially in contexts of rapid structural change, such as global health emergencies, natural disasters, or major social disruptions.

## Conclusion

In conclusion, loss to follow-up (LTFU) during the period of telemedicine implementation was associated with initiating care after the introduction of telemedicine and with having the first consultation in the Emergency Department. However, there was no evidence that telemedicine itself—as a mode of care—functioned as either a risk or protective factor for LTFU. Further research is needed to better characterize the profiles of older adults who seek psychiatric care through emergency services and to identify predictors of LTFU in this vulnerable population. As telemedicine becomes a stable and integral component of mental health care, its implementation should be closely monitored and continuously refined to maximize its potential benefits.

## Author contributions

**Conceptualization:** Paulo Ruiz-Grosso, Luis Macedo-Orrego, Sonia Zevallos-Bustamante.

**Data curation:** Paulo Ruiz-Grosso, Luis Macedo-Orrego, Sonia Zevallos-Bustamante.

**Formal analysis:** Paulo Ruiz-Grosso.

**Methodology:** Paulo Ruiz-Grosso.

**Project administration:** Luis Macedo-Orrego.

**Software:** Paulo Ruiz-Grosso.

**Supervision:** Sonia Zevallos-Bustamante.

**Writing – original draft:** Paulo Ruiz-Grosso, Luis Macedo-Orrego, Sonia Zevallos-Bustamante.

**Writing – review & editing:** Paulo Ruiz-Grosso, Luis Macedo-Orrego, Sonia Zevallos-Bustamante.

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
