## [Decision Letter · Decision Letter 0]

14 Jul 2025

PMEN-D-25-00222

Changes in Loss to Follow-Up Among Elderly Patients with Schizophrenia During the COVID-19 Pandemic: A Study from a Specialized Center in Peru

PLOS Mental Health

Dear Dr. Ruiz-Grosso,

Thank you for submitting your manuscript to PLOS Mental Health. After careful consideration, we feel that it has merit but does not fully meet PLOS Mental Health’s publication criteria as it currently stands. Therefore, we invite you to submit a revised version of the manuscript that addresses the points raised during the review process.

We look forward to receiving your revised manuscript.

Kind regards,

Lambert Zixin Li, Ph.D.

Academic Editor

PLOS Mental Health

Journal Requirements:

Additional Editor Comments (if provided):

Reviewers' comments:

Reviewer's Responses to Questions

**Comments to the Author**

1. Does this manuscript meet PLOS Mental Health’s publication criteria?

Reviewer #1: Partly

Reviewer #2: Yes

2. Has the statistical analysis been performed appropriately and rigorously?

Reviewer #1: Yes

Reviewer #2: Yes

3. Have the authors made all data underlying the findings in their manuscript fully available (please refer to the Data Availability Statement at the start of the manuscript PDF file)?

Reviewer #1: Yes

Reviewer #2: Yes

4. Is the manuscript presented in an intelligible fashion and written in standard English?

Reviewer #1: Yes

Reviewer #2: Yes

Reviewer #1: This retrospective cohort study addresses an important and understudied question regarding loss to follow-up (LTFU) among elderly patients with schizophrenia during the COVID-19 pandemic in Peru. By examining the survival function and predictors of LTFU in relation to telemedicine implementation and the type of initial contact with care (ambulatory vs. emergency), the authors offer policy-relevant insights for mental health continuity in resource-constrained settings. However, several methodological and reporting concerns limit the interpretability and generalizability of the findings. Substantive revisions are needed to strengthen the study’s rigor, transparency, and alignment with epidemiological best practices.

Major comments:

1. The use of a random sample from clinical records is not clearly justified, especially given the small final sample size (n=138). A consecutive sampling strategy—including all eligible patients within the specified period—would have minimized selection bias and enhanced the representativeness of the findings. The rationale for random sampling should be explicitly discussed, or the authors should consider re-analyzing the data using a consecutive sample.

2. The authors acknowledge that key variables violated the proportional hazards assumption. However, simply applying robust variance may not adequately address this issue. Alternative modeling approaches as sensitivity analyses—such as stratified Cox models, time-varying covariates, or accelerated failure time models—should be explored and justified. The current models may yield biased hazard estimates if the assumption violations are not properly accounted for.

3. The exposure variable “post-telemedicine” implementation is conflated with the broader historical context of the COVID-19 pandemic, which introduced numerous system-wide disruptions (e.g., service closures, medication shortages). These confounders may drive the observed associations. The authors should more explicitly discuss the temporal and contextual confounding and clarify whether “telemedicine” serves as a proxy for broader systemic change rather than an isolated exposure.

4. The operational definition of LTFU—absence of subsequent scheduled visits without documentation of discharge, transfer, or death—risks misclassification. Some patients may have died, recovered, or transitioned informally to other services without proper documentation. This limitation should be more thoroughly addressed in the discussion, including its potential to bias survival estimates.

Minor comments:

1. The use of Directed Acyclic Graphs (DAGs) is a strength, but they are currently relegated to the supplementary material. At least one DAG should be included in the main text or figure section to support model selection decisions.

2. Replace or complement Table 2 with Kaplan-Meier survival curves, which would offer more intuitive and interpretable comparisons across groups (e.g., by type of admission or telemedicine period).

3. The manuscript uses causal phrasing (e.g., “risk factor,” “effect,” “impact”) despite the observational nature of the study and limitations in confounding control. Consider rephrasing to reflect associations rather than causal inferences.

4. Minor grammatical issues are present throughout the manuscript. For example: “a group a higher risk” (Introduction). A thorough copyedit is recommended.

Reviewer #2: Some minor grammatical and typographical errors should be corrected to enhance clarity such as verb-subject agreement, article use.

Practical implications such as how mental health services can improve follow-up care through structured protocols or hybrid models are only briefly mentioned and could be expanded. Similarly, theoretical implications too.

Overall, the manuscript contributes new knowledge to the field of mental health

**Do you want your identity to be public for this peer review?** For information about this choice, including consent withdrawal, please see our Privacy Policy

Reviewer #1: No

Reviewer #2: **Yes: ** Charles Ganaprakasam

---

## [Decision Letter · Decision Letter 1]

7 Oct 2025

Changes in Loss to Follow-Up Among Elderly Patients with Schizophrenia During the COVID-19 Pandemic: A Study from a Specialized Center in Peru

PMEN-D-25-00222R1

Dear Dr. Ruiz-Grosso,

We are pleased to inform you that your manuscript 'Changes in Loss to Follow-Up Among Elderly Patients with Schizophrenia During the COVID-19 Pandemic: A Study from a Specialized Center in Peru' has been provisionally accepted for publication in PLOS Mental Health.

Best regards,

Lambert Zixin Li, Ph.D.

Academic Editor

PLOS Mental Health

Thank you for addressing the reviewers' comments.

Reviewer Comments (if any, and for reference):

Reviewer's Responses to Questions

**Comments to the Author**

Reviewer #1: All comments have been addressed

Reviewer #2: All comments have been addressed

publication criteria?

Reviewer #1: Yes

Reviewer #2: Yes

3. Has the statistical analysis been performed appropriately and rigorously?

Reviewer #1: Yes

Reviewer #2: Yes

4. Have the authors made all data underlying the findings in their manuscript fully available (please refer to the Data Availability Statement at the start of the manuscript PDF file)?

Reviewer #1: Yes

Reviewer #2: Yes

5. Is the manuscript presented in an intelligible fashion and written in standard English?

Reviewer #1: Yes

Reviewer #2: Yes

Reviewer #1: (No Response)

Reviewer #2: The manuscript is well-written, methodologically sound, and provides valuable insights into how telemedicine and patterns of service entry influence treatment continuity

**Do you want your identity to be public for this peer review?** For information about this choice, including consent withdrawal, please see our Privacy Policy

Reviewer #1: No

Reviewer #2: **Yes: ** Charles Ganaprakasam
